# Investigating a visceral measure of perceived physical attractiveness

Molly A. Bowdring[1¤], Michael A. Sayette[1], Kasey G. Creswell[2]*

1 Department of Psychology, University of Pittsburgh, Pittsburgh, PA, United States of America,
2 Department of Psychology, Carnegie Mellon University, Pittsburgh, PA, United States of America

¤ Current address: Stanford Prevention Research Center, Stanford University School of Medicine, Palo Alto, CA, United States of America
* kasey@andrew.cmu.edu

## Abstract

Perceptions of physical attractiveness are typically assessed using numeric rating scales. As with other visceral experiences, perceptions of physical attractiveness may benefit from multimodal measurement. Recently, we developed and validated a squeeze (dynamometer) method to evaluate two "visceral" states (hunger and cigarette craving). Here, we extend this approach to perceptions of physical attractiveness. Participants ($n = 33$) viewed a series of static facial images. Perceptions of physical attractiveness were assessed using the dynamometer, followed by a traditional rating scale ranging from 1 (*very unattractive*) to 10 (*very attractive*). Participants also reported desire to (a) interact with each individual they viewed in a future study and (b) become friends with each individual they viewed, using a Likert scale. Dynamometer-measured perceptions of physical attractiveness were significantly associated with traditional perceptions of physical attractiveness ratings and predicted both desire outcomes. Findings offer initial support for a visceral approach to perceptions of physical attractiveness that can complement traditional rating scales.

## Introduction

Physical attractiveness perceptions are affectively-charged visceral experiences, with attractiveness judgments made within a fraction of a second [1, 2]. Research on perceptions of physical attractiveness (PPA) has traditionally relied on verbal measures that require perceivers to translate their nonverbal perception experience into a numerical representation on a rating scale (e.g., 1: *very unattractive* to 10: *very attractive*) [3]. These ratings have led to a number of discoveries related to PPA, such as the tendency to ascribe favorable non-physical traits to attractive others and to treat attractive others more positively [3–5]. Nevertheless, exclusive reliance on verbal self-reports of PPA may be subject to some of the concerns associated with other affectively-charged "visceral" states, such as craving and sexual arousal [6, 7]. As we have noted, "these visceral states are thought to be inherently nonverbal, and participants may have difficulty translating these inner experiences into symbolic systems (such as numbers and language) required for traditional 'verbal' self-report rating scales" ([8], p. 598).

**Data Availability Statement:** The data underlying the results presented in the study are available

from: https://osf.io/gdwfu/?view_only=0296b93bd5904c74979c624ddde7d95b.

**Funding:** The Research Society on Alcoholism Graduate Student Small Grants Award to MAB, internal funds provided by the Department of Psychology at the University of Pittsburgh to MAB, an National Heart Lung and Blood Institute funded postdoctoral fellowship to MAB (5T32HL161270-02), and the U.S. National Institute on Alcohol Abuse and Alcoholism (Grant R01 AA015773) to MAS. The funders had no role in study design, data collection and analysis, decision to publish, or preparation of the manuscript.

**Competing interests:** The authors have declared that no competing interests exist.

We recently introduced a complementary nonverbal measure (squeezing a dynamometer, used to assess grip strength) to examine visceral states [8, 9]. For example, we have observed that dynamometer-measurements of both hunger and cigarette craving correlate with their verbal-measurement counterparts and, even predict eating behavior better than does the verbal hunger measurement [8, 9]. More recently, other labs have used a dynamometer to assess moments of insight (predictive of accurate responses on insight problem solving) and emotionality (predictive of a physiological measure of emotionality) [10, 11]. We sought to extend the set of visceral states amenable to nonverbal measurement. To this end, we investigated whether (1) nonverbally-measured PPA (via dynamometer) relates to verbally-measured PPA and (2) this measure predicts desire to interact and become friends with the person being perceived (target).

## Methods

### Participants

Participants were male social drinkers aged 21–28, recruited from 8/2019-3/2020. [Though we are disinclined to preference male recruitment, the focus of the larger study was on alcohol-related social rewards, which are stronger for males [12]. We selected males to enhance power to detect effects. The alcohol component of the study is unrelated to this report.] Participants had to have a nonromantic same-sex friend who would also participate. Participants were excluded if they denied fluency in English or had uncorrected visual impairment.

### Procedure

This report derives from a study approved by the University of Pittsburgh's IRB. During this two-session study, participants provided written informed consent, completed questionnaires, consumed either alcohol or no-alcohol control beverages with their friend, and completed multiple tasks related to person perception. See [13, 14] for details. The focus here is on a PPA task completed by participants separately after the session-two drinking period.

Participants viewed 16 static facial images extracted from video from a previous study (participants consented to use of their images; [15]). The images included four male and four female targets, each presented once with a positive emotional expression [Duchenne smile–action units 6 (cheek raiser)+12 (lip corner puller)+25 (lips part)] and once with a neutral expression (no emotion-relevant action units), defined by FACS coding [16]. Images appeared for 5-secs each. Participants viewed the images in three rounds to complete the: nonverbal PPA measure; verbal PPA measure; and measure of desire to interact with and be friends with the targets.

### Measures

**Nonverbal PPA.** Participants indicated how attractive they perceived targets by squeezing a dynamometer using standard procedures [8, 9]. Participants used their dominant hand to squeeze as long and as forcefully as necessary to express their attraction. To account for force and time, area under the curve was used as the nonverbal PPA measure [8, 9].

**Verbal PPA.** Participants self-reported their PPA using a scale of 1 (*very unattractive*) to 10 (*very attractive*) [17].

**Desire to interact and be friends.** Participants reported their desire to (a) interact with each target in a future study and (b) become friends with each target, using a scale of 1 (*not at all*) to 10 (*very much*) (modified from [18]).

**Analytic plan.** Mixed effects models included random intercepts for participants and targets. Likelihood ratio tests compared full models against models with nonverbal PPA removed. First, we tested the association between nonverbal PPA and verbal PPA. Next, we tested the association between nonverbal PPA and desire to interact (or be friends), then added verbal PPA. We controlled for target sex, target emotional expression, and drink condition in all analyses–with one exception reported in the final model, these covariates were not significant.

## Results

### Participants

Of the 36 participants, 33 had valid data on the PPA task. Ages ranged from 21–27. Participants were White ($n = 18$), Asian ($n = 12$), Black ($n = 2$), and Hispanic ($n = 1$). Thirty-one were heterosexual, 1 was gay, and 1 was bisexual.

### Nonverbal PPA associations with verbal PPA, and desire to interact and be friends with target

See Table 1 for variable means and Table 2 for associations. Nonverbal PPA was associated with verbal PPA. Nonverbal PPA was also associated with both desire to interact and desire to be friends. Verbal PPA also was related to desire to interact and to be friends. After controlling for verbal PPA, nonverbal PPA no longer was significantly linked to desire to interact or to be friends. Target sex was a significant covariate ($\beta = 0.19$, 95%CI: 0.03, 0.34, $p = .02$)–participants had greater desire to be friends with male targets.

## Discussion

This study evaluated a novel tool for assessing PPA. Our nonverbal, dynamometer-based PPA assessment was strongly associated with verbal PPA. Moreover, the dynamometer assessment was associated with desire to interact and to be friends with targets, offering initial support for its use as a complementary measure of PPA to verbal ratings.

The dynamometer responses did not explain additional variance beyond what was captured by verbal PPA. This is perhaps unsurprising, however, as desire to interact and be friends were also assessed with verbal measures. Prior work using the dynamometer has demonstrated its predictive validity with a behavioral outcome of eating [9]. Future PPA studies may examine whether dynamometer-based PPA predicts nonverbal outcomes, including actual approach behaviors in live social interactions [19], as nonverbal PPA might be more likely to outperform verbal PPA in these instances. Indeed, PPA is influenced by what the target can offer (e.g., romantic love, friendship, entertainment) and the environment in which perception occurs [20]. Just as craving for substances is heightened when the substance is accessible [21–23], the visceral experience of perceiving someone else as attractive may be heightened when that person is available for live interaction.

The ability to capture visceral states in real-time remains an important yet elusive research objective. While verbal self-report remains the gold standard, translating visceral states into

**Table 1. Means and standard deviations of verbal PPA and desire outcomes.**

|  | *Mean* | *SD* |
|---|---|---|
| **Verbal PPA** | 4.15 | 1.95 |
| **Desire to Interact** | 5.29 | 2.09 |
| **Desire to be Friends** | 5.25 | 2.24 |

**Table 2. Nonverbal PPA associations with verbal PPA, and desire to interact and be friends with the target.**

| | $\beta$ | 95% CI | $X^2$ (1, $N = 528$) | $p$-Value |
|---|---|---|---|---|
| **Associations with Verbal PPA** | | | | |
| Nonverbal PPA | 0.71 | 0.62, 0.81 | 160.18 | < .001 |
| **Associations with Desire to Interact** | | | | |
| *Model Without Verbal PPA* | | | | |
| Nonverbal PPA | 0.42 | 0.31, 0.53 | 50.36 | < .001 |
| *Model With Verbal PPA* | | | | |
| Nonverbal PPA | 0.07 | -0.04, 0.19 | 0.40 | .53 |
| Verbal PPA | 0.57 | 0.48, 0.66 | - | - |
| **Associations with Desire to be Friends** | | | | |
| *Model Without Verbal PPA* | | | | |
| Nonverbal PPA | 0.36 | 0.24, 0.47 | 34.65 | < .001 |
| *Model With Verbal PPA* | | | | |
| Nonverbal PPA | 0.04 | -0.08, 0.16 | 0.51 | .48 |
| Verbal PPA | 0.49 | 0.40, 0.59 | - | - |

quantitative metrics required for rating scales can be challenging. In some cases, verbalizing a rating may even distort the otherwise difficult-to-articulate state (see [9]). The development of a nonverbal dynamometer measure to complement traditional self-report ratings holds promise for advancing knowledge of visceral states. Together with studies of hunger, craving, emotionality, and insight [8–11], this study extends the utility of a dynamometer-measure to complement verbal PPA ratings. Further, the validity of dynamometer-measured PPA suggests exciting directions for future work assessing preference in academic and marketing settings.

## Author Contributions

**Conceptualization:** Molly A. Bowdring, Michael A. Sayette.

**Formal analysis:** Molly A. Bowdring.

**Funding acquisition:** Molly A. Bowdring, Michael A. Sayette.

**Investigation:** Molly A. Bowdring.

**Methodology:** Molly A. Bowdring, Michael A. Sayette, Kasey G. Creswell.

**Resources:** Michael A. Sayette.

**Supervision:** Michael A. Sayette.

**Writing – original draft:** Molly A. Bowdring.

**Writing – review & editing:** Molly A. Bowdring, Michael A. Sayette, Kasey G. Creswell.

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
