## [Decision Letter · Decision Letter 0]

12 Aug 2024

PONE-D-24-24619Investigating a Visceral Measure of Perceived Physical AttractivenessPLOS ONE

Dear Dr. Creswell,

Thank you for submitting your manuscript to PLOS ONE. After careful consideration, we feel that it has merit but does not fully meet PLOS ONE’s publication criteria as it currently stands. Therefore, we invite you to submit a revised version of the manuscript that addresses the points raised during the review process.

We look forward to receiving your revised manuscript.

Kind regards,

Jayaprakash Narayana Kolla

Academic Editor

PLOS ONE

Journal Requirements:

2. Please expand the acronym “NHLBI” (as indicated in your financial disclosure) so that it states the name of your funders in full.

Additional Editor Comments:

Comments:

1.The authors report on a method where squeezing a device would complement verbal judgments on perceived physical attractiveness of facial images.

2.In the methods, the authors state that the alcohol component of the study does not play a role in this report. But a few lines further down, they say that a PPA task was completed after the second drinking period. This sounds like a contradiction to me. Please explain.

3.The images to assess included 4 male and 4 females. This looks like not enough to get a relevant result. Please explain.

4. How is the squeezing standardized? I assume, different people have different squeezing strength but want to express a similar thing. Whereas the same strength might mean something different to different individuals. Base on this assumption, I also can not see how squeezing strength and verbal expression can be associated.

5. The authors have conveyed the study and its findings clearly and succinctly. This is an important research in social psychology that involves physiological underpinnings. It helps understand and further the study of interpersonal attraction.

Reviewers' comments:

Reviewer's Responses to Questions

**Comments to the Author**

1. Is the manuscript technically sound, and do the data support the conclusions?

Reviewer #1: Partly

Reviewer #2: Yes

2. Has the statistical analysis been performed appropriately and rigorously? 

Reviewer #1: I Don't Know

Reviewer #2: Yes

3. Have the authors made all data underlying the findings in their manuscript fully available?

Reviewer #1: Yes

Reviewer #2: Yes

4. Is the manuscript presented in an intelligible fashion and written in standard English?

Reviewer #1: Yes

Reviewer #2: Yes

5. Review Comments to the Author

Reviewer #1: The authors report on a method where squeezing a device would complement verbal judgments on perceived physical attractiveness of facial images.

In the methods, the authors state that the alcohol component of the study does not play a role in this report. But a few lines further down, they say that a PPA task was completed after the second drinking period. This sounds like a contradiction to me. Please explain.

The images to assess included 4 male and 4 females. This looks like not enough to get a relevant result. Please explain.

How is the squeezing standardized? I assume, different people have different squeezing strength but want to express a similar thing. Whereas the same strength might mean something different to different individuals. Base on this assumption, I also can not see how squeezing strength and verbal expression can be associated.

Reviewer #2: The authors have conveyed the study and its findings clearly and succinctly. This is an important research in social psychology that involves physiological underpinnings. It helps understand and further the study of interpersonal attraction.

6. PLOS authors have the option to publish the peer review history of their article (what does this mean?). If published, this will include your full peer review and any attached files.

Reviewer #1: No

Reviewer #2: No

---

## [Author Response · Author response to Decision Letter 0]

15 Aug 2024

Jayaprakash Narayana Kolla August 15, 2024

Academic Editor

PLOS ONE 

Dear Dr. Kolla,

We very much appreciate your invitation to revise and resubmit for publication consideration our manuscript entitled, “Investigating a Visceral Measure of Perceived Physical Attractiveness” [PONE-D-24-24619]. This letter details our point-by-point responses to the comments provided by the reviewers, who offered valuable recommendations for our manuscript. 

Journal Requirements

1. Please ensure that your manuscript meets PLOS ONE's style requirements.

We have addressed the style requirements. 

2. Please expand the acronym “NHLBI” (as indicated in your financial disclosure) so that it states the name of your funders in full. This information should be included in your cover letter. 

We have expanded the acronym in the financial disclosure and now note funding in the cover letter.

This information is included in the methods section (p.4): “This report derives from a study approved by the University of Pittsburgh’s IRB. During this two-session study, participants provided written informed consent…”

Reviewer 1

1. In the methods, the authors state that the alcohol component of the study does not play a role in this report. But a few lines further down, they say that a PPA task was completed after the second drinking period. This sounds like a contradiction to me. Please explain.

While the larger study was focused on drink condition differences, the goal of the task that is the focus of the present report was to compare two measures of attractiveness perceptions (dynamometer and verbal) irrespective of drink condition. As noted on lines 95-96, we control for drink condition in all analyses and it was not significant. We do not elaborate further on the drink condition, but instead refer readers to other publications for details on the drink manipulation if of interest (line 68).

2. The images to assess included 4 male and 4 females. This looks like not enough to get a relevant result. Please explain.

This is a within person comparison of two attractiveness measures (dynamometer and verbal). To control for the impact of target on measure variance, both measures are based on the same set of 16 facial images (8 targets). There is not reason to think more targets would alter the findings, as they would be presented in both assessment modes.

3. How is the squeezing standardized? I assume, different people have different squeezing strength but want to express a similar thing. Whereas the same strength might mean something different to different individuals. Base on this assumption, I also can not see how squeezing strength and verbal expression can be associated.

As noted above, this is a within person comparison. We assessed the relation between squeezing and verbal ratings within each target, within each participant. Thus, each participant serves as their own control. The random intercept in the mixed effects models accounted for potential variation across participants.

Reviewer 2

We were please Reviewer 2 found our paper to report on “important research in social psychology” that will “further the study of interpersonal attraction.” This reviewer had no requested revisions.

In closing, it is our hope that we have adequately addressed the recommendations offered to us and we look forward to hearing from you.

Sincerely,

Molly A. Bowdring, PhD

---

## [Decision Letter · Decision Letter 1]

16 Sep 2024

Investigating a visceral measure of perceived physical attractiveness

PONE-D-24-24619R1

Dear Dr. Creswell,

We’re pleased to inform you that your manuscript has been judged scientifically suitable for publication and will be formally accepted for publication once it meets all outstanding technical requirements.

Kind regards,

Jayaprakash Narayana Kolla

Academic Editor

PLOS ONE

Additional Editor Comments (optional):

Reviewers' comments:

Reviewer's Responses to Questions

**Comments to the Author**

1. If the authors have adequately addressed your comments raised in a previous round of review and you feel that this manuscript is now acceptable for publication, you may indicate that here to bypass the “Comments to the Author” section, enter your conflict of interest statement in the “Confidential to Editor” section, and submit your "Accept" recommendation.

Reviewer #2: All comments have been addressed

2. Is the manuscript technically sound, and do the data support the conclusions?

Reviewer #2: Yes

3. Has the statistical analysis been performed appropriately and rigorously? 

Reviewer #2: Yes

4. Have the authors made all data underlying the findings in their manuscript fully available?

Reviewer #2: Yes

5. Is the manuscript presented in an intelligible fashion and written in standard English?

Reviewer #2: Yes

6. Review Comments to the Author

Reviewer #2: I do not have further comments. The manuscript looks alright to publish now; the authors have adequately addressed the questions raised in the previous review.

7. PLOS authors have the option to publish the peer review history of their article (what does this mean?). If published, this will include your full peer review and any attached files.

Reviewer #2: No

---

## [Editor Report · Acceptance letter]

27 Sep 2024

PONE-D-24-24619R1 

PLOS ONE

Dear Dr. Creswell, 

I'm pleased to inform you that your manuscript has been deemed suitable for publication in PLOS ONE. Congratulations! Your manuscript is now being handed over to our production team.

Kind regards, 

on behalf of

Dr. Jayaprakash Narayana Kolla 

Academic Editor

PLOS ONE